# Alteration of Lipid Metabolism in Patients with IPF and Its Association with Disease Severity and Prognosis: A Case–Control Study

**DOI:** 10.3390/ijms26125790

**Published:** 2025-06-17

**Authors:** Paola Faverio, Paola Rebora, Giovanni Franco, Anna Amato, Nicole Corti, Katya Cattaneo, Simona Spiti, Umberto Zanini, Alessandro Maloberti, Cristina Giannattasio, Fabrizio Luppi, Valerio Leoni

**Affiliations:** 1U.O.C. di Pneumologia, Fondazione IRCCS San Gerardo dei Tintori, 20900 Monza, Italy; g.franco@campus.unimib.it (G.F.); u.zanini@campus.unimib.it (U.Z.); fabrizio.luppi@unimib.it (F.L.); 2School of Medicine and Surgery, Università degli Studi di Milano-Bicocca, 20100 Milan, Italy; nicole.corti@asst-brianza.it (N.C.); katya.cattaneo@asst-brianza.it (K.C.); alessandro.maloberti@unimib.it (A.M.); cristina.giannattasio@unimib.it (C.G.); valerio.leoni@unimib.it (V.L.); 3School of Medicine and Surgery, Bicocca Center of Bioinformatics, Biostatistics and Bioimaging (B4 Center), Università degli Studi di Milano-Bicocca, 20100 Milan, Italy; paola.rebora@unimib.it (P.R.); a.amato29@campus.unimib.it (A.A.); 4Unit of Clinical Epidemiology and Biostatistics, Fondazione IRCCS San Gerardo dei Tintori, 20900 Monza, Italy; 5Laboratory of Clinical Pathology, Hospital Pio XI of Desio, ASST-Brianza, 20832 Desio, Italy; simona.spiti@asst-brianza.it; 6Cardiology IV, ASST Grande Ospedale Metropolitano Niguarda, 20100 Milan, Italy

**Keywords:** idiopathic pulmonary fibrosis, lipidomics, mitochondria, peroxisome, prognosis

## Abstract

The pathogenesis of idiopathic pulmonary fibrosis (IPF) involves complex interactions between epithelial, mesenchymal, immune, and endothelial cells, often aggravated by lipid metabolism dysfunction, mitochondrial, and peroxisomal abnormalities. Changes in lipid metabolism may drive fibrotic processes, suggesting the potential of lipid biomarkers for disease monitoring. We compared here the cholesterol metabolism and very-long-chain fatty acid profiles of patients with IPF with healthy controls. The IPF patients’ lipidic profiles were also evaluated according to disease severity and progression rate. This prospective, observational study involved 50 IPF patients at disease diagnosis before antifibrotic treatment initiation and 50 age- and gender-matched healthy controls. Using a serum lipidomic profile, we focused on cholesterol synthesis, mitochondrial and peroxisomal markers, inflammatory lipids, and oxidative stress markers. Disease severity was evaluated using the Gender-Age-Physiology (GAP) index, while the prognosis was assessed by classifying patients as rapid or slow progressors based on a 24-month follow-up. IPF patients exhibited lower levels of cholesterol synthesis precursors (e.g., lathosterol), mitochondrial oxysterols, and inflammatory mediators (e.g., arachidonic acid) compared to controls. Reduced levels of these biomarkers were also associated with higher disease severity and rapid disease progression. Conversely, some peroxisomal markers (e.g., brassidic acid and nervonic acid) showed altered trends depending on disease severity. Our findings indicate that patients with IPF, compared to healthy controls, may show lipidomic alterations, particularly a reduction in cholesterol precursors and docosahexaenoic acids, which are also associated with IPF severity and progression. While preliminary, this study suggests lipidomics to be a promising tool to stratify IPF severity and prognosis.

## 1. Introduction

Idiopathic pulmonary fibrosis (IPF) is a chronic life-threatening fibrosing interstitial pneumonia of unknown cause [1]. The pathogenesis of IPF involves a complex interaction between various cell types and signaling pathways. Repeated injury to alveolar epithelial cells, often in the presence of predisposing factors, can lead to metabolic dysfunction, cellular senescence, abnormal epithelial activation, and impaired repair mechanisms. Dysfunctional epithelial cells interact with mesenchymal, immune, and endothelial cells through multiple signaling pathways, ultimately driving fibroblast and myofibroblast activation [2]. Recent studies have also explored the role of lipid metabolism, as well as mitochondrial and peroxisomal function, in the development of IPF [3,4,5,6].

Alterations in lipid metabolism, encompassing fatty acids, cholesterol, and arachidonic acid metabolites, play significant roles in the pathogenesis of IPF. These lipid mediators influence cellular processes such as endothelial reticulum (ER) stress, apoptosis, and pro-fibrotic signaling, thereby contributing to the onset and progression of the disease. In particular, saturated fatty acids, such as palmitic acid (C16:0), can accumulate in lung tissues during IPF and induce ER stress by disrupting lipid homeostasis. Prolonged or unresolved ER stress can trigger apoptotic pathways, contributing to epithelial cell death and fibrotic progression [7,8].

Cholesterol metabolism is also altered in IPF. Elevated cholesterol levels have been observed in bronchoalveolar lavage fluid from IPF patients, and cholesterol derivatives, such as 25-hydroxycholesterol, can promote myofibroblast differentiation and extracellular matrix deposition through TGF-β and NF-κB signaling pathways [9]. Arachidonic acid and its metabolites, including prostaglandins and leukotrienes, are involved in inflammatory responses and tissue remodeling. In IPF, dysregulated arachidonic acid metabolism can lead to the production of pro-inflammatory and pro-fibrotic mediators, exacerbating lung injury and fibrosis [3,4].

Alteration of peroxisome and mitochondrial activity may in turn favor pulmonary inflammation, cellular senescence and profibrotic response [5,6]. Lipidomic analyses may contribute to the exploration of biomarkers of disease severity and prognosis and for new therapeutic targets.

The purpose of this study is to compare the lipid metabolism of patients with IPF at disease diagnosis with healthy controls and to evaluate differences in the lipidomic profile among IPF patients with different disease severity and progression rates.

## 2. Results

Patients with IPF were predominantly male (78%) and former smokers (72%), with a median age of 74 years [interquartile range: 70–76]. Key demographic and clinical characteristics are summarized in Table 1. The most frequent comorbidities in both the IPF and control groups were arterial hypertension and dyslipidemia. Statin therapy was reported in 21 IPF patients and 8 control subjects, while ezetimibe was used by 1 IPF patient and 3 controls. Based on the GAP index, 27 patients were classified as stage I, 17 as stage II, and 6 as stage III.

The distribution of lipid markers in IPF patients—stratified by GAP stage—and in control subjects is presented in Appendix A. Figure 1 illustrates the distribution of cholesterol synthesis precursors, 27OHC (a mitochondrial-derived oxysterol), arachidonic acid and DHA, which are precursors of inflammatory mediators such as prostaglandins and leukotrienes. These markers were significantly lower in IPF patients, particularly in those with more advanced disease (GAP stages II and III), and higher in control subjects. The oxysterols originated by cholesterol non-enzymatic oxidation (7βOHC, 7KC, α-EpoxyC, β-EpoxyC, triol) and markers of oxidative stress were the lowest in healthy controls, growing from patients with mild disease to the highest levels in those with severe disease (Appendix A). Serum very-long-chain fatty acids (VLCFAs), markers of peroxisomal function, demonstrated variable trends. Certain VLCFAs—such as brassidic acid, erucic acid, and behenic acid—were elevated in healthy controls and less severe IPF patients, and decreased in patients with more advanced disease. Conversely, the levels of nervonic acid, lignoceric acid, cerotic acid, and ultra-long-chain fatty acids, which all together play a role in peroxisomal catabolism, were lower in healthy subjects and those with mild disease but increased in patients with more severe IPF (Figure 2).

Comparative analysis of IPF patients and control subjects (Table 2) confirmed lower levels of lanosterol, 27OHC, inflammatory markers, and brassidic, erucic, and behenic acid in the IPF group, even after adjusting for sex, age, statin use, and multiple testing. Furthermore, lanosterol levels were significantly lower in patients with more severe IPF compared to those with less severe disease (OR = 0.085, 95% CI: 0.014–0.287).

In contrast, nervonic acid, lignoceric acid, cerotic acid and ultra-long-chain fatty acids were significantly elevated in IPF patients. Notably, for each standard deviation increase in these metabolites, the odds of having IPF (versus being a control) were at least fourfold higher, while holding other variables constant.

Among the 50 patients with IPF, the median follow-up duration was 38 months; 5 patients had a follow-up period of less than 24 months and were excluded from the present analysis. Of the remaining 45 patients, 9 were classified as rapid progressors (including 5 deaths and 1 acute exacerbation), while 36 were categorized as slow progressors. During the study period, 46 patients received antifibrotic therapy; 4 either declined or discontinued treatment due to intolerance. Among those who continued therapy, 26 received nintedanib (19 slow progressors, 4 rapid progressors, 3 lost to follow-up), and 20 were treated with pirfenidone (15 slow progressors, 5 rapid progressors).

The trends observed in relation to disease severity were consistent when stratified by disease progression. Specifically, levels of cholesterol synthesis precursors, 27OHC, arachidonic acid, and especially DHA were lower in rapid progressors compared to slow progressors, as shown in Table 3. In contrast, levels of VLCFAs were generally comparable between the two groups.

## 3. Discussion

In this study we explored the lipidic profile in IPF patients at the time of diagnosis (before the initiation of antifibrotic treatment), comparing it with age- and sex-matched non-smoking controls, and evaluated their association with disease severity and prognosis. Several markers of lipid metabolism, including cholesterol synthesis precursors, mitochondrial oxysterols, and inflammatory mediator precursors such as arachidonic acid and DHA, were higher in healthy controls compared to patients with IPF.

Additionally, our study highlighted that reductions in cholesterol synthesis precursors, particularly lanosterol, and inflammatory mediators, particularly DHA, at IPF diagnosis were associated with greater disease severity at baseline (as assessed by the GAP score) and with more rapid disease progression over the 24-month follow-up.

Regarding cholesterol metabolism, there was a reduction in the cholesterol precursors (lathosterol, lanosterol, and desmosterol) consistent with the studies reporting an impairment of the cholesterol synthesis in acute and chronic inflammatory diseases [11,12]. Also, it has been shown that hypocholesterolemia and malnutrition negatively impairs clinical outcomes in IPF patients, especially during acute disease exacerbations [4].

Furthermore, there is growing evidence that unhealthy aging, along with mitochondrial and peroxisomal dysfunction, plays a role in the initiation and progression of pulmonary fibrosis [5,6,13]. Therefore, lipid metabolites may serve as diagnostic and prognostic markers of the disease.

In our cohort, we found that the oxysterol 27OHC, which originates in the inner mitochondria by CYP27A1, was reduced in patients with IPF, suggesting an impairment of the mitochondrial function as observed in other pathological conditions [14,15].

Also, the reduction in brassidic acid, erucic acid, and behenic acid (C22 and C22:1) and the increase in longer fatty acids (nervonic acid, lignoceric acid, cerotic acid, C24, C24:1 and C26) is consistent with the finding of peroxisomal impairment [16,17]. An increase was also found in other metabolic conditions with inflammation and cellular damage [14,15].

In particular, we observed a dramatic increase in the concentration of nervonic acid in IPF patients compared to controls. Nervonic acid is known for its anti-inflammatory and neuroprotective properties. Although most studies have focused on neurological disorders, emerging evidence suggests a potential role for nervonic acid in chronic respiratory diseases, particularly through the modulation of inflammation and macrophage activation [18]. Given the very limited research available on this topic, further studies are required.

Furthermore, this dramatic increase in patients with IPF is common also to lignoceric acid, cerotic acid, and ultra-long-chain fatty acids. Lignoceric acid (C24:0) is found in various cellular lipids, including ceramides and sphingomyelins, which are involved in regulating membrane fluidity and stability. Alterations in the metabolism of these lipids can affect inflammatory responses and pulmonary fibrosis. A study conducted in patients with chronic obstructive pulmonary disease (COPD) in 2013 evaluated these mechanisms and found that the expression of lipids from the sphingolipid pathway was higher in the sputum of smokers with COPD compared with smokers without COPD and showed high correlations with lower lung function and inflammation in sputum [19].

In regard to cerotic acid and ultra-long-chain fatty acids, current evidence remains insufficient to establish a direct role in chronic lung diseases.

The increase in oxidative stress biomarkers in patients with IPF, particularly in those with more severe disease, is consistent with the literature [20,21,22]. Several studies have demonstrated significantly elevated systemic oxidative stress levels in IPF patients compared to healthy controls. These elevated oxidative stress levels inversely correlate with pulmonary function parameters and positively correlate with dyspnea severity. Longitudinal data indicate that these markers increase over time in untreated patients, correlating with worsening lung function and disease progression.

We have to acknowledge also some limitations: (i) the small sample size, which reduces our power to detect important associations, in particular when restricted to patient data in the subanalysis on disease severity and prognosis; (ii) lipidomic analyses were performed solely on blood samples, with no samples from the airways or lungs (e.g., bronchoalveolar lavage); (iii) biological samples were collected only at the time of IPF diagnosis, so we cannot draw the temporal trends of the metabolites, particularly in relation to the introduction of antifibrotic therapies; and (iv) lipid-lowering therapies could impact on lipidomic evaluation. However, only a minority of patients took statins or ezetimibe limiting this risk, and we adjusted for statin use in multivariable models. Furthermore, a sensitivity analysis excluding patients on lipid-lowering therapies present the same results of the main analysis, as shown in Appendix A.

A better characterization of disease severity and prognosis, through metabolomic signatures, could improve the categorization of IPF patients for therapeutic and follow-up purposes. Future studies should include larger sample sizes and validation cohorts to confirm our preliminary results.

## 4. Materials and Methods

In this prospective, observational study, we recruited 50 consecutive patients with an IPF diagnosis from the outpatient specialist clinic of San Gerardo Hospital, Monza, Italy, and 50 controls, recruited among hospital volunteers, matched for age (±5 years) and gender, between 2019 and 2022. Patients were eligible for inclusion if they received an IPF diagnosis according to the ATS/ERS/JRS/LATS 2018 guidelines and had not yet started antifibrotic treatment^1^. Exclusion criteria (for both patients and controls) were as follows: active smoking, presence of atrial fibrillation or flutter, limb amputation and/or severe peripheral vasculopathy, and oxygen therapy at rest [23].

For both patients and controls, blood samples were collected after an overnight fasting in the morning according with standard guidelines [24], by standard venepuncture in vacutainer tubes (Greiner) avoiding blood stasis. After clothing, serum was separated by centrifugation at 3.500× *g* × 5 min. Samples were than kept frozen under −80 °C until the time of the mass spectrometry analysis. For patients with IPF, blood sampling was performed immediately after the diagnosis and before starting antifibrotic treatment. Data collected included comorbidities and concomitant medications, and, for patients with IPF, disease severity at baseline (evaluated through the Gender-Age-Physiology (GAP) index [25]).

IPF patients were followed-up according to clinical practice every 6 to 9 months, and mortality, onset of acute exacerbations of IPF, and functional worsening were recorded. Follow-up was updated as July 2024. Patients with IPF were divided into rapid or slow progressors after a follow-up of 24 months. Patients were classified as rapid disease progressors during the 24-month follow-up in the case of all-cause mortality, the onset of at least one episode of acute exacerbation, or at least two criteria among the following: a forced vital capacity (FVC) decrease ≥ 10%, a deterioration of chronic respiratory symptoms, or a worsening of fibrotic alterations on high-resolution CT scan compared to baseline [26,27].

The lipids analyzed were as follows: (i) the precursor sterols lathosterol, desmosterol, and lanosterol as markers of the cholesterol synthesis; (ii) 27-hydroxycholesterol (27OHC), formed from cholesterol by the mitochondrial cholesterol 27-hydroxylase (CYP27A1) as a marker of mitochondrial function; (iii) arachidonic and docosahexaenoic –DHA- acids, polyunsaturated fatty acid (PUFA) precursors of inflammatory mediators; (iv) very-long-chain fatty acids (VLFA, such as brassidic acid -n-9 trans docosenoic acid-, erucic acid -n-9, cis docosenoic acid-, behenic acid -C_21_H_43_COOH-, nervonic acid -C_24_H_46_O_2_-, lignoceric acid -C_23_H_47_COOH-, cerotic acid -C_26_H_52_O_2_-, and ultra-long-chain fatty acid -C26:1-), markers of peroxisome function; and (v) the oxysterols 7β-hydroxycholesterol (7βOHC), 7α-hydroxycholesterol (7αOHC), 7-ketocholesterol (7KC), 5,6α-epoxycholesterol (α-EpoxyC), 5,6β-epoxycholesterol (β-EpoxyC), and triol products of oxysterols autoxidation as oxidative stress markers [28].

### 4.1. Fatty Acid, Sterol, and Oxysterol Quantification by Gas Chromatography–Mass Spectrometry

To a screw capped vial sealed with a Teflon septum, 200 μL of plasma was added together with structural homologous internal standards: pentadecanoic 50 µg, heptadecanoic acid 100 µg, nonadecanoic acid 5 µg, heneicosanoic acid 2500 ng, tricosanoic acid 1000 ng, epicoprostanol 100 µg, d7-lathosterol (D7-latho) 500 ng, d6-campesterol (d6-campe) 250 ng, d7-β-sitosterol, d6-lanosterol 50 ng, 7α-hydroxycholesterol-d7 (D7-7αOHC) 50 ng, 7β-hydroxycholesterol-d7 (d7-7βOHC) 50 ng, 7-ketocholesterol-d7 (d7-7KC) 50 ng, 5α,6α-epoxycholestanol-d7 (d7-a-epoxyC) 20 ng, 5ß,6ß-epoxycholestanol-d7 (d7-b-epoxyC) 50 ng, cholestane-3ß,5α,6ß-triol-d7 (d7-triol) 50 ng, 24(R/S)-hydroxycholesterol-d7(D7-24OHC) 50 ng, 25-hydroxycholesterol-d6 (d6-25OHC) 50 ng, and 27-hydroxycholesterol-d6(d6-27OHC) 50 ng (Avanti Polar Lipids Inc., Alabaster, AL, USA), 50 μL of butylatedhydroxytoluene (BHT, 5 g/L), 50 μL of K3-EDTA (10 g/L) to prevent auto-oxidation, ethanol, and KOH 1M. Each vial was flushed with argon for 10 min to remove air. Hydrolysis was carried at room temperature.

Sterols and oxysterols were collected by liquid-to-liquid extraction with 5 mL of hexane. Fatty acids were collected after correction of pH (<3) with HCl by liquid-to-liquid extraction with 4 mL of hexane twice. The organic solvents were evaporated under a gentle stream of argon and converted into trimethylsilyl ethers with bis(trimethylsilyl)trifluoroacetamide with 1% trimethylchlorosilane (Pierce).

Gas chromatography–mass spectrometry (GC–MS) analysis was performed on a Agilent 5973 Mass spectrometer connected to a GC 6890 Agilent equipped with an HP-5MS columns (30 m × 0.32 mm id × 0.25 mm film; Agilent, Santa Clara, CA, USA), and injection was performed in splitless mode using helium (1 mL/min) as carrier gas. Injection was carried at 250 °C with a flow rate of 20 mL/min. The transfer line temperature was 290 °C. The filament temperature was set at 150 °C and quadrupole temperature at 220 °C, according to the manufacturer’s instructions.

The temperature program was as follows: initial temperature 190 °C held for 1 min, followed by a linear ramp of 10 °C/min to 300 °C, and then held for 8 min.

Mass spectrometric data were acquired in selected ion monitoring mode (OTMSi-ethers) at *m*/*z* = 355 for nonadecanoic (C19), *m*/*z* = 383 for heneicosanoic (C21), *m*/*z* 411 for tricosanoic (C23), *m*/*z* = 361 for arachidonic (C20:4), *m*/*z* = 359 for eicosapentaenoic (C20:5), *m*/*z* = 363 for meadic (C20:3), *m*/*z* = 365 for eicosadienoic (C20:2), *m*/*z* = 365 for eicosenoic (C20:1), *m*/*z* = 367 for eicosanoic (C20), *m*/*z* = 215 for decosohexenoic (c22:6), *m*/*z* = 395 for brassidic and eurcic acid (C22:1), *m*/*z* = 397 for behenic (C22), *m*/*z* = 423 for nervonic (C24:1), *m*/*z* = 425 for lignoceric (C24), *m*/*z* = 451 for C26:1 and *m*/*z* = 453 for cerotic (C26) acid, at *m*/*z* = 463 for 7α-hydroxycholesterol-d7, *m*/*z* = 456 for 7α-hydroxycholesterol, at *m*/*z* = 463 for 7β-hydroxycholesterol- d7, *m*/*z* = 456 for 7β-hydroxycholesterol, *m*/*z* = 479 for 7-ketocholesterol- d7 and *m*/*z* = 472 for 7-ketocholesterol, *m*/*z* = 131 for 25-hydroxycholesterol and *m*/*z* 137 for 25-hydroxycholesterol-d6, *m*/*z* 413 for D7-24S-hydroxycholesterol and 24S-hydroxycholesteorl, *m*/*z* = 462 for 27-hydroxycholesterol—d6, *m*/*z* = 456 for 27-hydroxycholesterol at *m*/*z* = 370 for epicoprostanol, and *m*/*z* = 368 for cholesterol.

Peak integration was performed manually, and oxysterols were quantified from selected ion monitoring analysis against internal standards using standard curves for the listed sterol fatty acids, sterols and oxysterols [29].

### 4.2. Statistical Methods

Data are described by frequencies and percentages and by median and interquartiles range as appropriate. Box-plot and violin plots were used to depict the distribution of the markers in IPF patients, stratified by GAP index (I, II–III) and controls.

Multivariable logistic models have been used to compare the markers values among IPF patients and controls adjusting for predefined variables (sex, age, and statin use). Given that the markers had different scales, and to facilitate interpretation, we centered and scaled the marker values before fitting the model (subtracted their mean and divided by their standard deviation). This resulted in coefficients representing the change in the response variable for one standard deviation change in the marker, rather than a one-unit change in the original variable. When needed, a mixed bias-reducing adjusted score equations (Firth’s bias reduction method [10]) was used to account for quasi-complete separation and sparse data. The same approach was used among/within IPF patients to compare marker values in patients with GAP index II, III versus I. In order to account for the multiplicity of the statistical test, a conservative adjustment procedure has been used using Bonferroni (we divided the significance level 0.05 by the number of tests).

Markers values among IPF patients classified as rapid or slow progressors over the follow-up of at least 24 months have been described by quartiles and compared by the Mann–Whitney U test.

This study received Ethics Committee approval and was registered on www.clinicaltrials.gov (NCT04177251—registration date 21 October 2019). All patients and controls agreed to participate in the study and signed a written informed consent.

## 5. Conclusions

Our findings reveal alterations in lipid metabolism in IPF patients compared to healthy controls, and among patients with IPF stratified by disease severity and prognosis. On the one hand, these alterations suggest that certain metabolites, particularly lanosterol, a cholesterol precursor, may serve as biomarkers for assessing disease severity. Conversely, the marked elevation of VLCFAs—particularly nervonic, lignoceric, cerotic acid and other ultra-long-chain fatty acids—represents a novel finding in patients with IPF, and more broadly in individuals with chronic respiratory diseases. The clinical relevance of this lipid profile alteration remains to be elucidated and warrants further investigation.

## Figures and Tables

**Figure 1 ijms-26-05790-f001:**
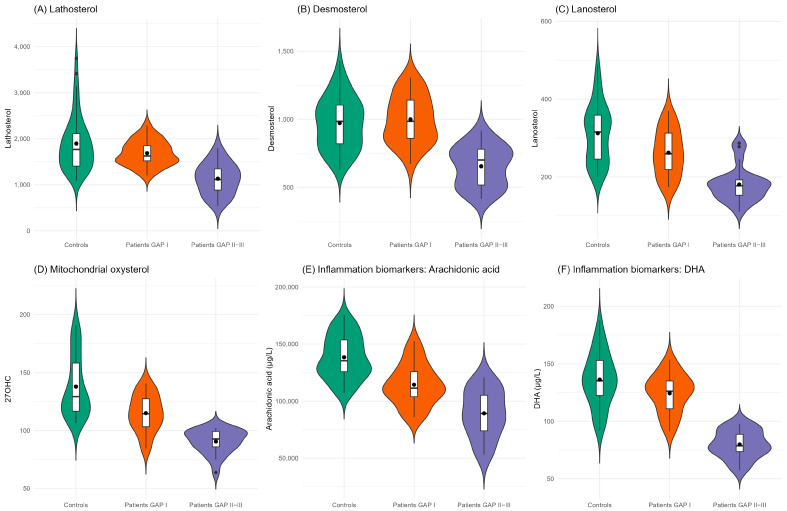
Distribution of metabolites by box-plot and violin plot among controls (green) and IPF patients with GAP index I (orange) and II–III (purple/violet). Cholesterol synthesis precursors (lathosterol, desmosterol, lanosterol) are reported in the top row and mitochondrial oxysterol (27OHC) and inflammation biomarkers (arachidonic acid and docosahexaenoic acid) in the bottom row. The violin plots display probability density estimates of the data calculated using kernel density estimation with Gaussian kernels and Silverman’s rule for bandwidth selection. The width of each violin at any given point represents the estimated probability density at that value. The embedded boxplots indicate statistical dispersion with the following components: the central box spans from the first quartile to the third quartile, with a horizontal line indicating the median. Mean values are represented by solid points, calculated as the arithmetic mean of all non-missing values for each category.

**Figure 2 ijms-26-05790-f002:**
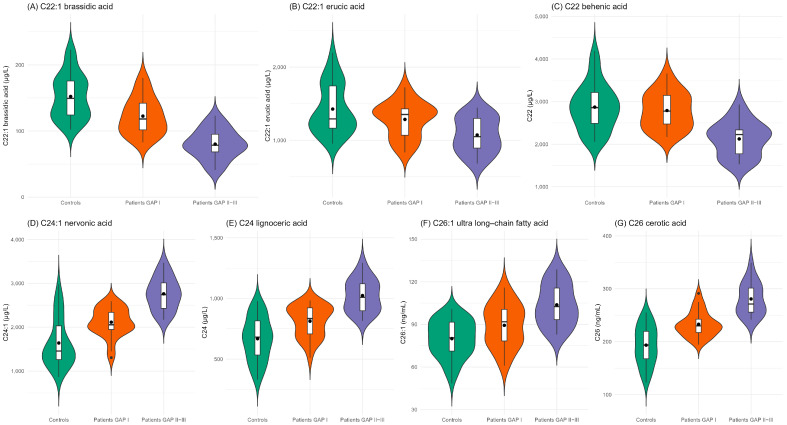
Distribution of very-long-chain fatty acids by box-plot and violin plot among controls (green) and IPF patients with GAP index I (orange) and II–III (purple/violet). The violin plots display probability density estimates of the data calculated using kernel density estimation with Gaussian kernels and Silverman’s rule for bandwidth selection. The width of each violin at any given point represents the estimated probability density at that value. The embedded boxplots indicate statistical dispersion with the following components: the central box spans from the first quartile to the third quartile, with a horizontal line indicating the median. Mean values are represented by solid points, calculated as the arithmetic mean of all non-missing values for each category.

**Table 1 ijms-26-05790-t001:** Baseline characteristics of patients with idiopathic pulmonary fibrosis (IPF), stratified by the Gender-Age-Physiology (GAP) index, and controls.

	Controls (*n* = 50)	Patients—GAP Index I (*n* = 27)	Patients—GAP Index II–III (*n* = 23)
**Demographics**			
**Age (years), median [I–III quartiles]**	72 [68, 76]	72 [69, 75]	75 [72, 77]
	**N (%)**	**N (%)**	**N (%)**
**Males (%)**	39 (78)	18 (67)	21 (91)
**Never smoker ***	22 (45)	8 (30)	6 (26)
**Comorbidities**			
**Arterial hypertension**	25 (50)	11 (41)	13 (57)
**Dyslipidemia**	13 (26)	10 (37)	11 (48)
**Diabetes type 2**	7 (14)	5 (19)	7 (30)
**Obesity ****	1 (2)	2 (8)	1 (5)
**Valvular heart disease ***	2 (4)	0 (0)	2 (9)
**Coronary artery disease ***	4 (8)	1 (4)	9 (39)
**Prior myocardial infarction ***	3 (6)	1 (4)	5 (22)
**Peripheral arterial disease**	0 (0)	1 (4)	3 (13)
**Pharmacological Therapy**			
**Statins *****	8 (16)	10 (37)	11 (48)
**Ezetimibe**	3 (6)	1 (4)	0 (0)
**Pulmonary function tests at baseline**			
**FVC%, median [I–III quartiles]. ****	-	97 [88, 107]	70 [65, 77]
**DLCO%, median [I–III quartiles]. ******	-	5.0 [4.4, 5.8]	3.1 [2.7, 3.7]

FVC% = forced vital capacity percentage of predicted value; DLCO% = diffusing capacity for carbon monoxide percentage of predicted value; * 1 missing; ** 2 missing; *** 11 missing; **** 3 missing.

**Table 2 ijms-26-05790-t002:** Multivariable logistic models results comparing IPF patients versus controls (N = 86) and less severe (GAP stage I) vs. more severe patients (GAP stage II and III) (*n* = 47), adjusted for age, sex and statin use. As markers have been centered and scaled, OR represents the change in the response variable for a standard deviation change in the marker.

Characteristic	IPF pts vs. Controls (N = 86)			GAP II, III vs. I (N = 47)		
	OR ^1^	95% CI ^1^	*p*-Value	OR ^1^	95% CI ^1^	*p*-Value
	**Cholesterol synthesis precursors**					
**Lathosterol**	0.336	0.152, 0.655	0.003	0.047	0.004, 0.218	0.002
**Desmosterol**	0.653	0.388, 1.065	0.095	0.017	0.000, 0.132	0.003
**Lanosterol**	**0.281**	**0.139, 0.513**	**<0.001 ***	**0.085**	**0.014, 0.287**	**<0.001 ***
	**Mitochondrial oxysterol**					
**27-hydroxycholesterol**	**0.103**	**0.030, 0.267**	**<0.001 ***	0.03	0.002, 0.183	0.003
	**Inflammation biomarkers**					
**Arachidonic acid**	**0.045**	**0.009, 0.142**	**<0.001 ***	0.192	0.053, 0.506	0.003
**Docosahexaenoic acid ^**	**0.154**	**0.067, 0.354**	**<0.001 ***	0.023	0.002, 0.265	0.003
	**Very-long-chain fatty acids**					
**C22:1 brassidic acid**	**0.183**	**0.072, 0.375**	**<0.001 ***	0.037	0.002, 0.200	0.002
**C22:1 erucic acid**	0.391	0.206, 0.678	0.002	0.396	0.166, 0.833	0.021
**C22 behenic acid**	0.576	0.330, 0.957	0.04	0.09	0.013, 0.308	0.002
**C24:1 nervonic acid**	**4.525**	**2.380, 9.828**	**<0.001 ***	35.686	6.217, 575.303	0.001
**C24 lignoceric acid**	**5.253**	**2.649, 12.242**	**<0.001 ***	13.599	3.557, 110.217	0.002
**C26:1 ultra-long-chain fatty acid**	**12.228**	**4.584, 45.908**	**<0.001 ***	12.284	3.528, 83.602	0.001
**C26 cerotic acid**	**4.013**	**2.191, 8.364**	**<0.001 ***	3.682	1.632, 10.398	0.005

^1^ OR = Odds ratio for a standard deviation increase in the marker, CI = confidence interval; * significant after Bonferroni adjustment (*p*-value < (0.05/26)); ^ mixed bias-reducing adjusted score equations [10].

**Table 3 ijms-26-05790-t003:** Distribution of metabolites according to slow and rapid progressors in a 24-month follow-up. * Among the 36 slow progressors, 3 did not have marker measurements.

	Slow Progressor (*n* = 33 *)	Rapid Progressor (*n* = 9)
	Median [I–III Quartiles]	Median [I–III Quartiles]
**Cholesterol synthesis precursors**		
**Lathosterol (µg/L)**	1542.16 [1296.72, 1803.48]	1050.96 [979.52, 1344.20]
**Desmosterol (µg/L)**	858.72 [738.28, 985.84]	779.96 [572.88, 834.28]
**Lanosterol (µg/L)**	232.84 [177.04, 283.64]	188.84 [164.76, 245.96]
**Mitochondrial oxysterol**		
**27-hydroxycholesterol (µg/L)**	104.84 [88.64, 121.56]	99.16 [97.00, 102.00]
**Inflammation biomarkers**		
**Arachidonic acid (µg/L)**	107,250 [89,925, 116,160]	103,545 [89,175, 113,190]
**D** **ocosahexaenoic acid (µg/L)**	108.69 [91.44, 132.99]	76.92 [73.65, 97.98]
**Very-long-chain fatty acids**		
**C22:1 brassidic acid (µg/L)**	107.04 [91.00, 124.36]	83.44 [81.44, 95.16]
**C22:1 erucic acid (µg/L)**	1183.52 [941.40, 1392.40]	1276.08 [1011.12, 1344.68]
**C22 behenic acid (µg/L)**	2492.88 [2186.76, 2894.08]	2336.96 [2273.60, 2565.36]
**C24:1 nervonic acid (µg/L)**	2335.20 [2041.08, 2679.48]	2424.44 [2297.80, 2428.36]
**C24 lignoceric acid (µg/L)**	918.22 [800.82, 955.14]	897.27 [848.14, 1058.16]
**C26:1 ultra-long-chain fatty acid (µg/L)**	95.56 [81.16, 104.56]	93.30 [83.25, 100.46]
**C26 cerotic acid (µg/L)**	242.02 [225.78, 261.10]	265.54 [255.65, 271.20]

## Data Availability

The original contributions presented in this study are included in the article/Appendix A. Further inquiries can be directed to the corresponding author by email at paola.faverio@unimib.it.

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
