# Peer review of "Alteration of Lipid Metabolism in Patients with IPF and Its Association with Disease Severity and Prognosis: A Case–Control Study"

_ijms, 2025, doi:10.3390/ijms26125790_

Round 1
Reviewer 1 Report
Comments and Suggestions for Authors
Introduction is short which is not enough to cover background and abojective of the study. Extend it.
Under "Statins" for controls: "8 (21%)" is incorrect. If 8 out of 50 controls use statins, the percentage should be 16%, not 21%.
Nervonic acid (C24:1) shows an OR of 4.525 (95% CI: 2.380–9.828) for IPF vs. controls. This dramatic increase requires stronger biological justification in the Discussion.
Brassidic acid (C22:1 trans) and erucic acid (C22:1 cis) trends conflict: Brassidic acid is lower in IPF (OR = 0.083), while erucic acid is also lower (OR = 0.391). The text states both are elevated in controls but decreased in severe IPF. Clarify whether this refers to severity or case-control comparisons.
Lower arachidonic acid in IPF is attributed to "increased conversion to inflammatory mediators," but reduced substrate availability could also dampen inflammation. This contradiction needs resolution so cite evidence linking low levels to upregulated enzymatic activity.
Small cohort (n=50 IPF patients) limits statistical power, especially in subgroup analyses (e.g., GAP III: n=6). The authors acknowledge this, but validation in a larger cohort is critical for future work.
Statin use (21/50 IPF patients) may influence lipid profiles. While adjusted for in models, residual confounding is possible. A sensitivity analysis excluding statin users is mentioned but not shown.
Main conclusion part is missing in manuscript.
References are not accrording to MDPI Style.
Author Response
Reviewer 1
Comments and Suggestions for Authors
R1.C1
Introduction is short which is not enough to cover background and abojective of the study. Extend it.
R1.R1
We agree with the Reviewer and extended the content of the Introduction.
R1.C2
Under "Statins" for controls: "8 (21%)" is incorrect. If 8 out of 50 controls use statins, the percentage should be 16%, not 21%.
R1.R2
We thank the Reviewer and corrected the text accordingly.
R1.C3
Nervonic acid (C24:1) shows an OR of 4.525 (95% CI: 2.380–9.828) for IPF vs. controls. This dramatic increase requires stronger biological justification in the Discussion.
R1.R3
We thank the Reviewer for his/her comment. Nervonic acid is known for its anti-inflammatory and neuroprotective properties. Although most studies have focused on neurological disorders, emerging evidence suggests a potential role for nervonic acid in chronic respiratory diseases, particularly through the modulation of inflammation and macrophage activation.(Front Immunol. 2024 Dec 11:15:1405020.doi: 10.3389/fimmu.2024.1405020) Given the very limited research available on this topic, further studies are required.
We added the prior paragraph in the text.
Furthermore, since this dramatic increase is common also to lignoceric acid, cerotic acid and ultra long-chain fatty acids, we also added the following comment on the role of these metabolites in the Discussion: “Lignoceric acid (C24:0) is found in various cellular lipids, including ceramides and sphingomyelins, which are involved in regulating membrane fluidity and stability. Alterations in the metabolism of these lipids can affect inflammatory responses and pulmonary fibrosis. A study conducted in patients with Chronic Obstructive Pulmonary Disease (COPD) in 2013 evaluated these mechanisms and found that the expression of lipids from the sphingolipid pathway was higher in the sputum of smokers with COPD compared with smokers without COPD and showed high correlations with lower lung function and inflammation in sputum.[18]
In regards to cerotic acid and ultra long-chain fatty acids, current evidence remains insufficient to establish a direct role in chronic lung diseases.”
R1.C4
Brassidic acid (C22:1 trans) and erucic acid (C22:1 cis) trends conflict: Brassidic acid is lower in IPF (OR = 0.083), while erucic acid is also lower (OR = 0.391). The text states both are elevated in controls but decreased in severe IPF. Clarify whether this refers to severity or case-control comparisons.
R1.R4
From the descriptive analysis, there is a downward trend for both brassidic acid (C22:1 trans) and erucic acid (C22:1 cis). Both metabolites are higher in the control groups followed by less severe IPF patients and then more severe IPF, which is also observed in the logistic regression analysis.
R1.C5
Lower arachidonic acid in IPF is attributed to "increased conversion to inflammatory mediators," but reduced substrate availability could also dampen inflammation. This contradiction needs resolution so cite evidence linking low levels to upregulated enzymatic activity.
R1.R5
We agree with the Reviewer and in the absence of strong evidence linking low levels of arachidonic acid to upregulated enzymatic activity in patients with IPF, we deleted the speculation.
R1.C6
Small cohort (n=50 IPF patients) limits statistical power, especially in subgroup analyses (e.g., GAP III: n=6). The authors acknowledge this, but validation in a larger cohort is critical for future work.
R1.R6
We thank the Reviewer for the comment. We emphasized at the end of the Discussion that “Future studies should include larger sample sizes and validation cohorts to confirm our preliminary results.”
R1.C7
Statin use (21/50 IPF patients) may influence lipid profiles. While adjusted for in models, residual confounding is possible. A sensitivity analysis excluding statin users is mentioned but not shown.
R1.R7
We added the results of the multivariable logistic models results excluding persons on lipid-lowering therapies (statins or ezetimibe) in Supplementary Table 2.
R1.C8
Main conclusion part is missing in manuscript.
R1.R8
We thank the Reviewer for the comment. We added the paragraph Conclusion.
R1.C9
References are not accrording to MDPI Style
R1.R9
We amended as requested.
Reviewer 2 Report
Comments and Suggestions for Authors
Faverio and colleagues present their study examining the lipid profile in the plasma of IPF patients at the time of diagnosis and compare it to matched healthy controls. They also go on to assess the lipid profile at the time of diagnosis compared disease severity and progression. Of particular interest, they found that inflammatory markers such as arachidonic acid and DHA were reduced in IPF and related to disease severity. Some of these findings seems counterintuitive and as such make us think more deeply about the pathogenic mechanisms driving disease.
I enjoyed reading this manuscript. Attention to some minor points below would enhance the article.
- Abstract/introduction line 6 – The patients lipidic profile should be “patient’s”
- Reference 1 is not relevant to the statement, it is a review about lipids and IPF, not a review about IPF in general, which would be more appropriate. Also, (relevant to all references) when referencing a review it is helpful to the reader to state “(reviewed in (1))”.
- Reference 2 – same comment as for 1.
- Reference 3 – this article does not mention mitochondria (as stated).
- This sentence” Lipids, including fatty acids, cholesterol and its precursors and arachidonic acid me-tabolites are involved in the onset and progression of IPF by inducing endoplasmic retic-ulum stress, promoting cell apoptosis, and enhancing the expression of pro-fibrotic bi-omarkers2.” Does not make sense as written because all of these things are also all critical to lung homeostasis – more specific information would be helpful.
- Last paragraph of the introduction seems to be missing “The” purpose of this study… and the full stop is missing at the end.
- The blood sampling method could have more detail, e.g. tube type, process.
- Table 1; Is DLCO% meant to be DLCO%/va?
- The meaning of VLCFA is not explained.
- It might be better if the Supplemental table is included in the main text. Also note typographical error in that table “5,6b-aepoxy (µg/L)”
- Table 2 – it is confusing that (ug/L) is shown here, could that be omitted?
- At a glance It looks like the data (or label) for C26 and C26:1 might be swapped in the tables or figure.
- Nervonic acid is misspelled (Nervoic) in Table 2 and Table 3.
- Third last paragraph of discussion – typographical error – “restring”
Author Response
Reviewer 2
Comments and Suggestions for Authors
Faverio and colleagues present their study examining the lipid profile in the plasma of IPF patients at the time of diagnosis and compare it to matched healthy controls. They also go on to assess the lipid profile at the time of diagnosis compared disease severity and progression. Of particular interest, they found that inflammatory markers such as arachidonic acid and DHA were reduced in IPF and related to disease severity. Some of these findings seems counterintuitive and as such make us think more deeply about the pathogenic mechanisms driving disease.
I enjoyed reading this manuscript. Attention to some minor points below would enhance the article.
R2.C1
Abstract/introduction line 6 – The patients lipidic profile should be “patient’s”
R2.R1
We changed the text as suggested.
R2.C2
Reference 1 is not relevant to the statement, it is a review about lipids and IPF, not a review about IPF in general, which would be more appropriate. Also, (relevant to all references) when referencing a review it is helpful to the reader to state “(reviewed in (1))”.
R2.R2
We changed reference 1 as suggested with “Raghu G, Remy-Jardin M, Meyers JL, Richeldi L, Ryerson CJ, Lederer DJ, et al. Diagnosis of idiopathic pulmonary fibrosis: an official ATS/ERS/JRS/ALAT clinical practice guideline. Am J Respir Crit Care Med. 2018;198:e44–e68.”
R2.C3
Reference 2 – same comment as for 1.
R2.R3
We changed reference 2 as suggested with “Wolters PJ, Collard HR, Jones KD. Pathogenesis of idiopathic pulmonary fibrosis. Annu Rev Pathol. 2014;9:157-79.doi: 10.1146/annurev-pathol-012513-104706”
R2.C4
Reference 3 – this article does not mention mitochondria (as stated).
R2.R4
We changed reference 3 as suggested with reference 6 “Schuliga, M.; Pechkovsky, D. V.; Read, J.; Waters, D. W.; Blokland, K. E. C.; Reid, A. T.; Hogaboam, C. M.; Khalil, N.; Burgess, J. K.; Prêle, C. M.; Mutsaers, S. E.; Jaffar, J.; Westall, G.; Grainge, C.; Knight, D. A. Mitochondrial Dysfunction Contributes to the Senescent Phenotype of IPF Lung Fibroblasts. J Cell Mol Med 2018, 22 (12), 5847–5861. https://doi.org/10.1111/jcmm.13855.”
R2.C5
This sentence” Lipids, including fatty acids, cholesterol and its precursors and arachidonic acid me-tabolites are involved in the onset and progression of IPF by inducing endoplasmic retic-ulum stress, promoting cell apoptosis, and enhancing the expression of pro-fibrotic bi-omarkers2.” Does not make sense as written because all of these things are also all critical to lung homeostasis – more specific information would be helpful.
R2.R5
We thank the Reviewer for the opportunity to deepen this point. We changed the sentence and added the following sentence:
“alterations in lipid metabolism, encompassing fatty acids, cholesterol, and arachidonic acid metabolites, play significant roles in the pathogenesis of IPF. These lipid mediators influence cellular processes such as endothelial reticulum (ER) stress, apoptosis, and pro-fibrotic signaling, thereby contributing to the onset and progression of the disease. In particular, saturated fatty acids, such as palmitic acid (C16:0), can accumulate in lung tissues during IPF and induce ER stress by disrupting lipid homeostasis. Prolonged or unresolved ER stress can trigger apoptotic pathways, contributing to epithelial cell death and fibrotic progression.7,8
Cholesterol metabolism is also altered in IPF. Elevated cholesterol levels have been observed in bronchoalveolar lavage fluid from IPF patients, and cholesterol derivatives, such as 25-hydroxycholesterol, can promote myofibroblast differentiation and extracellular matrix deposition through TGF-β and NF-κB signaling pathways.9 Arachidonic acid and its metabolites, including prostaglandins and leukotrienes, are involved in inflammatory responses and tissue remodeling. In IPF, dysregulated arachidonic acid metabolism can lead to the production of pro-inflammatory and pro-fibrotic mediators, exacerbating lung injury and fibrosis.”
R2.C6
Last paragraph of the introduction seems to be missing “The” purpose of this study… and the full stop is missing at the end.
R2.R6
We thank the Reviewer for his/her corrections and change the text accordingly.
R2.C7
The blood sampling method could have more detail, e.g. tube type, process.
R2.R7
We added the following sentence in the Methods: “For both patients and controls, blood samples were collected after an overnight fasting in the morning according with standard guidelines (A. Tonkin, High-density lipoprotein cholesterol and treatment guidelines, Am. J. Cardiol. 12A (2001) 41N–44N), by standard venepuncture in vacutainer tubes (Greiner) avoiding blood stasis. After clothing, serum was separated by centrifugation at 3.500 x g x 5 min. Samples were than kept frozen under -80°C until the time of the mass spectrometry analysis.”
R2.C8
Table 1; Is DLCO% meant to be DLCO%/va?
R2.R8
No, DLCO% is DLCO percentage of predicted value. We changed the footnotes of Table 1 to clarify this point.
R2.C9
The meaning of VLCFA is not explained.
R2.R9
We thank the Reviewer for his/her corrections and change the text accordingly.
R2.C10
It might be better if the Supplemental table is included in the main text. Also note typographical error in that table “5,6b-aepoxy (µg/L)”
R2.R10
We thank the Reviewer for his/her suggestion. We left Supplementary Table 1 in the online supplements due to its dimensions.
R2.C11
Table 2 – it is confusing that (ug/L) is shown here, could that be omitted?
R2.R11
We thank the Reviewer for his/her corrections and change the text accordingly.
R2.C12
At a glance It looks like the data (or label) for C26 and C26:1 might be swapped in the tables or figure.
R2.R12
Thanks to point it out. Indeed the label was switched in Table 2 and the order was switched in Table 3. We corrected Table 2 and modified the order of the markers in Table 3 to be consistent with figures and supplementary tables.
R2.C13
Nervonic acid is misspelled (Nervoic) in Table 2 and Table 3.
R2.R13
We thank the Reviewer for his/her corrections and change the text accordingly.
R2.C14
Third last paragraph of discussion – typographical error – “restring”
R2.R14
We thank the Reviewer for his/her corrections and change the text accordingly.
Round 2
Reviewer 1 Report
Comments and Suggestions for Authors
Thanks for revision